# Detection and Real-Time Surgical Assessment of Colorectal Liver Metastases Using Near-Infrared Fluorescence Imaging during Laparoscopic and Robotic-Assisted Resections

**DOI:** 10.3390/cancers16091641

**Published:** 2024-04-24

**Authors:** Gaetano Piccolo, Matteo Barabino, Giorgio Ghilardi, Riccardo Masserano, Francesca Lecchi, Guglielmo Niccolò Piozzi, Paolo Pietro Bianchi

**Affiliations:** 1General Surgery Unit, Department of Health Sciences (DISS), San Paolo Hospital, University of Milan, 20142 Milano, Italy; gaetano.piccolo@asst-santipaolocarlo.it (G.P.); giorgio.ghilardi@unimi.it (G.G.); masseranoriccardo@gmail.com (R.M.); fra.lecchi@gmail.com (F.L.); bianchippt@gmail.com (P.P.B.); 2Department of Colorectal Surgery, Portsmouth Hospitals University NHS Trust, Portsmouth PO6 3LY, UK; guglielmopiozzi@gmail.com

**Keywords:** colorectal liver metastasis, robotic surgery, laparoscopic surgery, ICG, fluorescence, intraoperative ultrasound

## Abstract

**Simple Summary:**

Surgery still represents the gold standard for the treatment of colorectal liver metastases (CRLMs); thus, accurate evaluation of the number and location of nodules is crucial in order to achieve effective oncological results. Indocyanine green fluorescence (ICG) imaging, combined with intraoperative ultrasound, was revealed to be a valid and easily reproducible tool for this purpose. This study explored the use of ICG for the detection of tiny and superficial CRLMs during minimally invasive liver resection, using the integrity of the fluorescent rim around the lesion as a marker of radical resection (R0).

**Abstract:**

Background: The European Association of Endoscopic Surgery (EAES) recommends, with strong evidence, the use of indocyanine green (ICG) fluorescence imaging combined with intraoperative ultrasound (IOUS) to improve identification of superficial liver tumors. This study reports the use of ICG for the detection of colorectal liver metastases (CRLMs) during minimally invasive liver resection. Methods: A single-center consecutive series of minimally invasive (laparoscopic and robotic) hepatic resections for CRLMs was prospectively evaluated (April 2019 and October 2023). Results: A total of 25 patients were enrolled—11 undergoing laparoscopic and 14 undergoing robotic procedures. The median age was 65 (range 50–85) years. Fifty CRLMs were detected: twenty superficial, eight exophytic, seven shallow (<8 mm from the hepatic surface), and fifteen deep (>10 mm from the hepatic surface) lesions. The detection rates of CRLMs through preoperative imaging, laparoscopic ultrasound (LUS), ICG fluorescence, and combined modalities (ICG and LUS) were 88%, 90%, 68%, and 100%, respectively. ICG fluorescence staining allowed us to detect five small additional superficial lesions (not identified with other preoperative/intraoperative techniques). However, two lesions were false positive fluorescence accumulations. All rim fluorescence pattern lesions were CRLMs. ICG fluorescence was used as a real-time guide to assess surgical margins during parenchymal-sparing liver resections. All patients with integrity of the fluorescent rim around the CRLM displayed a radical resection during histopathological analysis. Four patients (8%) with a protruding rim or residual rim patterns had positive resection margins. Conclusions: ICG fluorescence imaging can be integrated with other conventional intraoperative imaging techniques to optimize intraoperative staging. Rim fluorescence proved to be a valid indicator of the resection margins: by removing the entire fluorescent area, a tumor-negative resection (R0) is achieved.

## 1. Introduction

Near-infrared fluorescence imaging (NIRF) has been adopted in hepatobiliary surgery for ischemic demarcation guiding parenchymal transection, cholangiography, and detection of small superficial tumors [1,2,3,4,5,6].

During anatomical liver resection, a precise delineation of the hepatic segments can be achieved by intravenous injection of indocyanine green (ICG, 2.5 mg bolus) after selective clamping of the segmental portal pedicle [7].

Recent studies have suggested that complete resection of all the ischemic liver tissue from the healthy liver is an independent predictor of long-term outcomes after hepatic surgery for malignant tumors. Moreover, during major hepatectomies, intrahepatic biliary anatomy can be evaluated using intraoperative fluorescent cholangiography with ICG to avoid severe complications such as biliary injury [8]. Another promising application of ICG fluorescence in liver surgery lies in the intraoperative detection of superficial liver tumors [9,10,11].

Accurate preoperative and intraoperative assessments of hepatic lesions are crucial steps of any scheduled parenchymal-sparing liver resections.

In the era of enhanced ultrasound-guided hepatectomy [12], three-dimensional (3D) anatomical reconstruction and ICG fluorescence seem to be valid tools to enhance the precision of liver resection [9]. For deep hepatic lesions, the three-dimensional (3D) reconstruction of radiological images allows the surgeon to assess the exact position of the tumor both before (virtual reality) and during the operation through immersive techniques (i.e., augmented reality) [13,14].

In minimally invasive settings, the lack of tactile feedback on the hepatic surface makes it challenging to detect subcapsular colorectal liver metastases (CRLMs). In these cases, ICG visual feedback might substitute the information given by palpation during open surgery [9]. The consensus statements of the European Association of Endoscopic Surgery (EAES) recommend, with strong evidence, the use of ICG fluorescence imaging to improve the detection of superficial liver tumors, combined with the use of laparoscopic ultrasound (LUS) to complement the accuracy of newly ICG-detected lesions [15].

This study reports the use of NIRF for the detection of CRLMs during minimally invasive liver resection and as a guide for the resection margins.

## 2. Material and Methods

### 2.1. Ethics

This study was conducted in accordance with the Declaration of Helsinki (6th revision, 2008) of the World Medical Association. The study was approved by our Institutional Research Board (ICG-CRLM230583), and informed consent for all surgical procedures was obtained from each patient.

### 2.2. Study Design

This study prospectively evaluated a consecutive series of minimally invasive (laparoscopic and robotic) hepatic resections for CRLMs performed between April 2019 and October 2023 at the San Paolo Hospital in Milan (Italy).

Inclusion criteria included (1) patients with low (1–3 peripheral lesions)-burden disease requiring minor/low-risk liver resections; (2) patients with intermediate (>3 peripherals lesions)-burden disease and with a concomitant single (≤3 cm) deep lesion suitable for microwave (MW) ablation.

Exclusion criteria included patients with high-burden disease requiring major liver resection, or patients with previous liver resections.

Before surgery, all patients were evaluated by a multidisciplinary tumor board (“Liver Group”) including oncologists, radiologists, and surgeons. Treatment strategy was decided based on both primary tumor factors (site, nodal status, and molecular characteristics such as KRAS mutation and microsatellite instability) and metastasis-related factors (tumor load, disease-free interval (DFI), and synchronicity). For patients with synchronous CRLMs, the simultaneous approach (resection of CRLMs together with the primary colorectal tumor) or the liver-first (reverse) strategy was chosen according to the hepatic tumor burden. The simultaneous approach was the preferred option for patients with a solitary metastasis requiring low-risk resection, while the reverse strategy was the preferred option for patients with multiple bilobar metastases. Before surgery, staging was achieved by sequential contrast-enhanced imaging studies, such as triple-phase helical computed tomography (CT) and contrast-enhanced magnetic resonance imaging (MRI). Based on the current literature, a standard low dose (10 mg) of ICG was injected the day before the scheduled surgery to all patients.

### 2.3. Intraoperative Staging

All patients underwent intraoperative staging with ultrasound and ICG fluorescence imaging to evaluate the number, the localization, and the fluorescence patterns of their hepatic lesions. ICG staining was also used to detect occult CRLMs (visual feedback instead of tactile feedback) (Figure 1) and for real-time assessment of the surgical margin status through the evaluation of the integrity of the fluorescent rim around each lesion (Figure 2).

### 2.4. Surgical Technique

All surgeries were performed by the same team of surgeons (MB and GP). A standard laparoscopic and robotic technique was adopted for all cases. The patients were placed in a supine reverse Trendelenburg position with legs apart. The operating surgeon or the laparoscopic assistant surgeon were standing between the legs, respectively, for the laparoscopic and robotic approaches. For the laparoscopic technique, two 5 mm ports and three 10 mm ports were used. Liver transection was performed using a laparoscopic ultrasonic dissector, or the Cavitron Ultrasonic Surgical Aspirator (CUSA). For the robotic technique, the patient cart was placed at the patient’s left side and the docking set-up was in the upper abdomen. Pneumoperitoneum was induced at 12 mmHg through a Veress needle inserted at the Palmer’s point. Trocars were placed as follows: optical port (8 mm) on the right mid-clavicular line along the transverse umbilical line; the other three operative trocars (8 mm) were placed in a straight line, two on the right and one on the left side (two right hands approach). Right trocars were used for the surgeon’s right hand (monopolar curved scissor or Maryland) and for the surgeon’s third hand (Cadiere forceps). The left trocar was used for the surgeon’s left hand (fenestrated bipolar forceps). Two additional assistant trocars (12 mm, 5 mm) were placed in the lower abdomen, between two robotic trocars. The assistant ports were used for retraction, suction, and extraction. Liver transection was performed through the double clamp crush technique using Maryland and bipolar forceps diathermy. Bile duct and vascular vessels were closed by the apposition of clips or stiches.

### 2.5. Statistical Analysis

Patients’ characteristics were summarized using basic descriptive statistics. Continuous variables were presented as median (interquartile range, IQR) or mean ± standard deviation accordingly, and compared using the Mann–Whitney U test. Statistical analysis was performed using IBM SPSS Statistics for Windows, version 22.0 (IBM Corp., Armonk, NY, USA). The STROBE statement principles for cohort studies were followed [16].

## 3. Results

### 3.1. Clinical Findings

A total of 25 consecutive cases were enrolled, including 11 laparoscopic and 14 robotic resections (Table 1). The median patient age was 65 years (range 50–85 years). Most patients were male (n = 15, 60%) and had a median body mass index (BMI) of 26.4 (range 18.5–36.7) kg/m^2^. Eight patients had concomitant liver disease (32%): four patients were affected by non-alcoholic fatty liver disease (NAFLD), two patients had chemotherapy-associated liver injury (CALI), and two patients had chronic hepatitis caused by hepatitis C viral infection (HCV).

### 3.2. Surgical Findings and ICG

Surgical procedures are shown in Table 2. Mean operative time was 250 min (range 180–300 min). No conversions to open surgery were reported. Median margin width was 10.5 mm (range 5–25 mm). An optimal ICG 15-min clearance retention rate (R15 < 10%) and an optimal postoperative ICG plasma disappearance rate (PDR) (<20%/minute) were found in 80% (n = 20) and in 76% of patients (n = 19), respectively. Primary tumor characteristics are shown in Table 3.

A significant subset of patients had left- or right-sided primary tumor localization—48% (n = 12) and 32% (n = 8), respectively. Most patients (64%) had a primary tumor at stage T3 (64%) or T4 (20%) with lymph node metastases (N1–N2). Most patients (60%) developed metachronous CRLMs and had two or more hepatic lesions. Most patients (88%) had unilobar distribution of the lesions, while only three patients (12%) had bilobar lesions. All patients with low-burden disease (1–3 peripheral lesions) requiring minor/low-risk liver resections underwent single-stage liver resections; among them, ten patients received neoadjuvant chemotherapy. Only three patients with synchronous CRLMs underwent up-front liver resection (“liver first approach”). Nine patients with intermediate-burden disease (1–5 peripheral lesions) and a concomitant single (≤3 cm) deep lesion, requiring major hepatic resection, underwent multiple low-risk liver resections and microwave (MW) ablation of the deep lesion. A total of 50 CRLMs were found: 20 superficial (surfacing at the liver’s Glissonian capsule), 8 exophytic, 7 shallow (<8 mm from the hepatic surface), and 15 deep (>10 mm from the hepatic surface) lesions.

Liver distribution and the total number of detected lesions via preoperative imaging (CT/MRI), intraoperative LUS, and ICG fluorescence in each segment (according to the Couinaud’s classification) are summarized in Table 4. The detection rates of CRLMs with preoperative imaging (abdominal CT/MRI), LUS, ICG fluorescence, and combined modalities (ICG and LUS) were 88%, 90%, 68%, and 100%, respectively. ICG fluorescence staining allowed us to detect five additional superficial lesions (not identified with other preoperative or intraoperative techniques), with diameters that were significantly smaller than those of additional CRLMs identified using LUS. However, two lesions were false positive fluorescence accumulations (Table 4).

The fluorescence of peripheral CRLMs on the Glissonian surface was classified into three patterns: rim fluorescence (n = 32/35, 91.4%), partial fluorescence (n = 2/35, 5.7%), and total fluorescence (n = 1/35, 2.8%). All lesions with the rim fluorescence pattern were CRLMs, while three lesions with intraoperative partial or total fluorescence patterns were found to be moderately or well-differentiated hepatocellular carcinoma (HCC) cells instead of CRLMs. In one of these cases, an additional superficial lesion was detected only with ICG (Figure 3).

In addition, NIRF has been adopted as a real-time guide to assess surgical margins during parenchymal-sparing liver resections. It is hypothesized that discontinuities of the fluorescent rim or the absence of overlaying healthy liver tissue around the CRLM could be considered as potential tumor-positive resection margins (Figure 4).

According to Achterberg et al. classification [17], three fluorescence patterns of CRLM resections were identified—(1) no fluorescent signal: no fluorescence signal produced by either the liver wound bed or the resection specimen; (2) protruding rim: a fluorescent rim protruding through the liver tissue of the resection specimen; (3) residual rim: a bright fluorescent signal in the wound bed of the resection specimen. All patients with the integrity of the fluorescent rim around the CRLM had a radical resection (resection margin with a distal clearance > 1 mm from the tumor, R0). Four patients (10.8%) with protruding rim or residual rim patterns had positive resection margins (resection margins < 1 mm from the tumor, R1) (Table 5).

### 3.3. Postoperative Outcomes

None of the patients required intraoperative blood transfusion. Postoperative complications were classified according to the Clavien–Dindo classification [18]. Grade II complications occurred in four (16%) patients: one patient developed a biliary leak (Grade A) responding to conservative management (drain in situ); one patient developed a fever and was treated only with antibiotics; one patient had an acute kidney injury due to intraoperative hypoperfusion, successfully treated with infusion therapy; and one patient developed chronic anemia requiring postoperative blood transfusion. The mean hospital stay was 6 days (range 4–8 days).

## 4. Discussion

This study reports the detection rates of CRLMs with preoperative imaging, LUS, NIRF, and combined modalities (ICG and LUS). NIRF allowed us to detect additional small superficial lesions that were not identified with other preoperative or intraoperative techniques.

Several studies have recently shown the utility of NIRF in liver surgery, suggesting that this technology is safe, feasible, and easy to use [19,20,21,22]. Reports were focused on the use of ICG to identify segmental hepatic regions and to obtain greater precision during parenchymal transection, especially during minimally invasive (laparoscopic or robotic) selective anatomical liver resection (segmentation) [23,24].

Two different techniques for visualization of the segmental hepatic anatomy with ICG have been described. In the positive staining technique, ICG is directly injected into a segmental portal branch [25], whilst in the negative staining technique, ICG is injected intravenously after selective clamping of the segmental portal pedicle [26,27,28].

Another application of NIRF is the detection of superficial hepatic lesions using an intravenous injection of ICG before surgery. Only small studies reported that the use of this technique for intraoperative identification of CRLMs produced excellent outcomes [17,19,29,30].

In a recent randomized controlled trial involving 32 patients for each group, the authors compared the numbers of CRLMs detected intraoperatively with and without the use of ICG fluorescence imaging [30]. The intraoperative identification of small (1–2 mm), previously unidentified CRLMs was significantly higher (3.03 SD, *p* = 1.58) in the ICG group compared to the non-ICG group (2.28 SD 1.35, *p* = 0.045). The authors also reported that the 1-year recurrence rate was significantly lower in the ICG group (47% vs. 19%, *p* = 0.017).

The usefulness of ICG fluorescence imaging was also confirmed by Piccolo et al. [29] in a recent review including 13 studies. The authors reported an average detection rate of CRLMs during minimally invasive liver resection of 79.03% (range 57.6–100%) with ICG, of 95.97% (range 93.3–100%) with LUS, and 100% with combined modalities (ICG and LUS).

In a prospective British study published in 2022 including 15 patients and thirty CRLMs, the authors reported that NIRF had higher sensitivity (93%) and specificity (100%) for the detection of small superficial lesions compared to the use of only white light/LUS [19]. As a result, twenty-three (77%) and twenty-seven (90%) resected CRLMs were detected with white light/LUS and NIRF, respectively. Among the seven (23%) lesions that were white light/LUS-negative, five were detected using ICG. NIRF was also useful in one patient with disappearing CRLMs after down-staging chemotherapy. In this case, only one out of three disappearing lesions was identified by white light/LUS, while all were ICG-positive [19]. Despite ICG fluorescence in all hepatic lesions, two CRLMs were found to have complete pathological responses, and only one of them contained viable cancer cells. The authors concluded that NIRF altered the operative plan in 43% of patients by confirming the benign nature of suspected CRLMs in two patients, and by localizing CRLMs not visualized by white light/LUS in four patients [19].

In the present study, the detection rates of CRLMs with preoperative imaging (abdominal CT/MRI), LUS, ICG fluorescence, and combined modalities (ICG and LUS) were 88%, 90%, 68%, and 100%, respectively. ICG fluorescence staining allowed us to detect five additional small, superficial lesions not identified with other preoperative or intraoperative techniques. However, two lesions were false positive. Interestingly, three lesions with non-typical rim fluorescence (intraoperative partial or total fluorescence patterns) were found to be moderately or well-differentiated hepatocellular carcinoma (HCC) cells instead of CRLMs.

ICG fluorescence imaging can also be used for the evaluation of surgical margins. Achterberg et al. [17] tested the utility of NIRF for real-time surgical margin assessment during laparoscopic and robotic resections of CRLMs in a pilot study. Seven out of eight patients with positive margins (<1 mm, R1) displayed a positive fluorescent signal from the specimen’s resection plane (protruding pattern). Of these, three patients also showed an NIRF signal in the wound bed of the residual liver (residual rim). Similar data have been reported in our prospective study, in which four patients (8%) with protruding or residual rim patterns had positive resection margins (resection margins <1 mm from the tumor, R1).

This study has some limitations. First, it is characterized by a limited sample size, which could limit the results. Second, the group is characterized by both laparoscopic and robotic approaches, which could potentially affect the results. Further studies comparing approaches and CRLM identifications should be performed. Third, this study was performed in a single centre with expertise in minimally invasive liver surgery and NIRF, which could impact the generalizability of the results. However, the study has strengths due to the prospective data collection and the fact that all procedures and imaging techniques were done homogenously between patients.

A large multicentric prospective cohort study has been initiated and is currently recruiting cases (MIMIC Trial, Dutch Trial Registry: NL 7674), with the aim of evaluating the rates of tumor-negative resection margins using ICG fluorescence as a real-time guide for CRLM resection. Disease recurrence and overall survival at six months and five years will also be analyzed.

## 5. Conclusions

ICG fluorescence imaging can be integrated with other conventional intraoperative imaging techniques (IOUS) to optimize intraoperative staging of superficial lesions. The most important drawback of this technique is the depth of fluorescence penetration (up to 8 mm from the liver surface). In this study, rim fluorescence was demonstrated to be a reliable indicator of the lesion margin; therefore, to obtain a tumor-negative resection (R0), the entire fluorescent area encircling the metastasis must be removed during surgery. Future research with larger-scale, multicentric studies may confirm that the use of ICG during minimally invasive CRLM surgery increases the rates of radical resection (R0), thereby increasing overall patient survival.

## Figures and Tables

**Figure 1 cancers-16-01641-f001:**
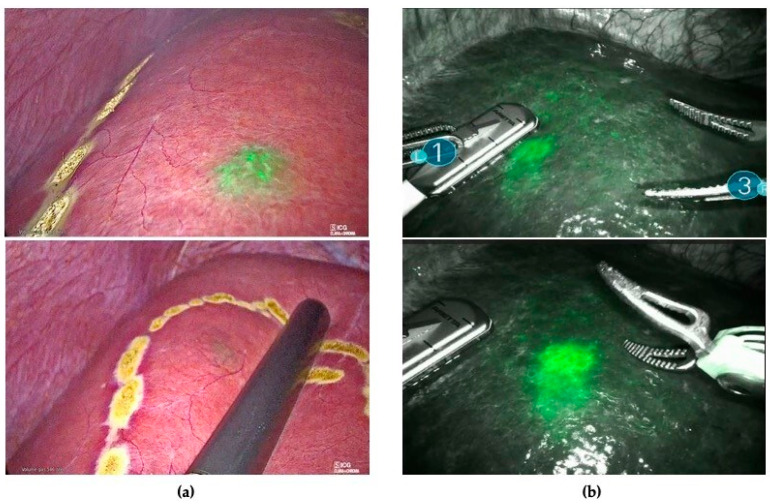
(**a**) The use of ICG as visual feedback to detect sub-capsular lesions not identified with ultrasound during laparoscopic resection. (**b**) ICG fluorescence combined with IOUS for nodule detection during robotic-assisted liver resection.

**Figure 2 cancers-16-01641-f002:**
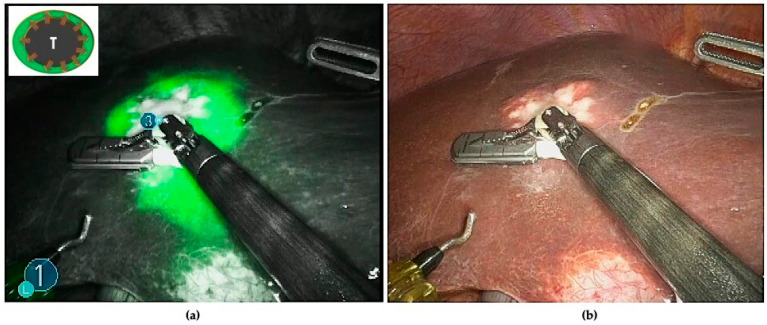
(**a**) The typical rim fluorescence of CRLMs is due to compressed hepatocytes, surrounding metastatic lesions, that have a decreased bile excretion ability. (**b**) The use of ICG combined with IOUS as a real-time guide to define the proper transection line.

**Figure 3 cancers-16-01641-f003:**
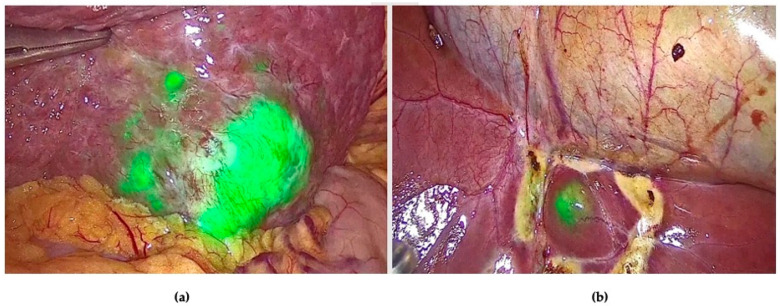
ICG fluorescence staining allowed us to detect: (**a**) the main nodule in S3 with a partial fluorescence pattern; (**b**) one additional smaller pericholecystic lesion with a total fluorescence pattern.

**Figure 4 cancers-16-01641-f004:**
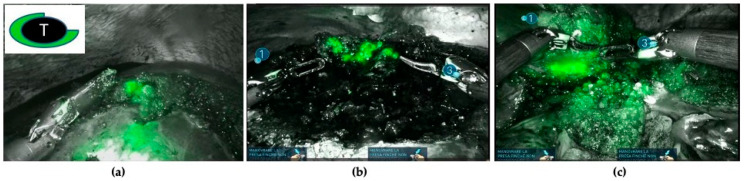
Three cases of positive resection margins are shown (resection margins < 1 mm from the tumor): (**a**) discontinuity of the rim fluorescence; (**b**) residual rim patterns (a bright fluorescent signal in the wound bed of the resection specimen); (**c**) protruding rim (a fluorescent area protruding through the liver tissue of the resected specimen).

**Table 1 cancers-16-01641-t001:** Patients’ characteristics. Variables are shown as median (range) or value (%).

	n = 25
**Age, years**	65 years (50–85)
**Sex**	
Male	15 (60%)
Female	10 (40%)
**BMI, kg/m^2^**	26.4 (18.5–36.7)
**ASA score**	
1–2	5 (20%)
3–4	20 (80%)
**Charlson Comorbidity Index (CCI)**	
Moderate CCI score (3–4)	18 (72%)
Severe CCI score (≥5)	12 (48%)
**Patient’s liver features**	
Normal	17 (68%)
NAFLD	4 (16%)
CALI	2 (8%)
Chronic hepatitis (HCV)	2 (8%)
**Limon Test R15 (%)**	
<10	20 (80%)
≥10	5 (20%)
**Limon Test PDR (%/min)**	
<20	19 (76%)
≥20	6 (24%)
**Number of tumors for patient**	
Single nodule	10 (40%)
Two or more nodules	15 (60%)
**Lobe distribution**	
Unilobar	22 (88%)
Bilobar	3 (12%)
**Tumor location of liver metastasis**	
Exophytic	8 (16%)
Superficial	20 (40%)
Shallow	7 (14%)
Deep	15/50 (30%)
**Neoadjuvant CHT**	
Yes	10 (40%)
No	15 (60%)
**Time of onset of liver metastasis**	
Synchronous	10 (40%)
Metachronous	15 (60%)

CALI = chemotherapy-associated liver injury; CCI = Charlson Comorbidity Index; CHT = chemotherapy; NAFLD = non-alcoholic fatty liver disease; PDR %/min = indocyanine green plasma disappearance rate; R15 % = indocyanine green 15 min clearance retention rate.

**Table 2 cancers-16-01641-t002:** Surgical procedures and intra- and postoperative outcomes of patients. Variables are shown as median (range) or value (%).

	n = 25
**Liver resection**	16 (64%)
**Liver resection + MW ablation of deep lesions**	9 (36%)
**Type of liver resection**	
Wedge resection	15 (60%)
Segmentectomy	6 (24%)
Left lateral sectionectomy	4 (16%)
**IWATE score**	
Low	2 (8%)
Intermediate	18 (72%)
Advanced	4 (16%)
Expert	2 (8%)
**Operative time, min**	250 (180–300)
**Estimated blood loss, mL**	250 (100–300)
Clavien–Dindo	
Grade II	4 (16%)
**Postoperative stay, days**	6 (4–8)
**Margin width, mm**	10.5 (5–25)

**Table 3 cancers-16-01641-t003:** Primary tumor characteristics. Variables are shown as value (%).

	n = 25
**Primary tumor location**	
Right colon	8 (32%)
Transverse colon	1 (4%)
Left colon	12 (48%)
Rectum	4 (16%)
**Tumor stage of primary tumor**	
T1	1 (4%)
T2	3 (12%)
T3	16 (64%)
T4	5 (20%)
**Tumor grade of primary tumor**	
G1	1 (4%)
G2	14 (56%)
G3	10 (40%)
**Nodal status of primary tumor**	
N0	9 (36%)
N1	9 (36%)
N2	7 (28%)

**Table 4 cancers-16-01641-t004:** Distribution of hepatic lesions detected by preoperative imaging, LUS, and ICG staining.

Segment	No (%) of Lesions Detected via Preoperative Imaging (CT/MR)	No (%) of Lesions Detected via LUS	No (%) of Lesions Detected via ICG	No (%) of Lesions Detected via LUS + ICG	Additional Lesions Detected Only via ICG (Yes/No, n)	False Positive Lesions (Yes/No, n)	Smallest Tumor Size Detected via ICG
**S2**	8/8 (100%)	8/8 (100%)	6/8 (75%)	8/8 (100%)	No	No	-
**S3**	6/6 (100%)	6/6 (100%)	4/6 (66.7%)	6/6 (100%)	No	No	-
**S4a**	4/5 (80%)	4/5 (80%)	5/5 (100%)	5/5 (100%)	Yes (n = 1)	Yes (n = 1)	6 mm
**S4b**	3/4 (75%)	3/4 (75%)	4/4 (100%)	4/4 (100%)	Yes (n = 1)	No	4 mm
**S5**	4/6 (66.7%)	4/6 (66.7%)	5/6 (83.3%)	6/6 (100%)	Yes (n = 2)	No	4 mm, 10 mm
**S6**	8/8 (100%)	8/8 (100%)	3/8 (37.5%)	8/8 (100%)	No	-	-
**S7**	9/10 (90%)	9/10 (90%)	6/10 (60%)	10/10 (100%)	Yes (n = 1)	Yes (n = 1)	8 mm
**S8**	2/3 (66.7%)	3/3 (100%)	1/3 (33.3%)	3/3 (100%)	No	-	-
**Tot**	44/50 (88%)	45/50 (90%)	34/50 (68%)	50/50 (100%)			

**Table 5 cancers-16-01641-t005:** ICG patterns of CRLM resections and radical resections.

	No Fluorescence Signal	Protruding Rim	Residual Rim	n (%)
**R0**	33	0	0	33/37 (89.2%)
**R1**	0	2	2	4/37 (10.8%)

R0 = resection margin with a distal clearance >1 mm from the tumor. R1 = resection margins < 1 mm from the tumor.

## Data Availability

The datasets used or analyzed during the current study are available from the corresponding author upon reasonable request.

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
