# Peer review of "Detection and Real-Time Surgical Assessment of Colorectal Liver Metastases Using Near-Infrared Fluorescence Imaging during Laparoscopic and Robotic-Assisted Resections"

_cancers, 2024, doi:10.3390/cancers16091641_

Round 1

Reviewer 1 Report

Comments and Suggestions for Authors

With interest I read your paper about NIRF during resection of CRLM.

I have some general comments:

What is the real novelty of your study? What is the main aspect? I mean, the usage of ICG is not new. You also cite others adequately in that context. Since you do not have data about overall survival or disease recurrence (addressed in a large study, I understood) and you have also false positive events, I miss a bit the novelty, the additional information above "it just works"...

For readers not familiar with minimal invasive settings, could you be so kind to explain a bit more what we see in the images. Label what is most important. Tell what are the numbers partially visible.  What is the T logo/label telling? Is there a scalebar/magnification value? etc.

Could you add a statement on how you judge the false positive lesions? All imaging techniques have a risk of such events, for sure. Reseasons are complex depending on technique, tissue(s), patient, contrast agents etc. What would you say (maybe on context to other techniques?), are these false positive things a problem? Or are they more like chance findings with a positiv effect possible because another type of malign/benign malformation is detected? Or is more likely to be adverse, that you would take to much intervention with the corresponding risks?

Small things:

Line 27 in Abstract: Fifry CRLMs ... Its a typo, right?

Line 115 in label of Fig 1: One (a) is too much, right?

Line 152: You are stating n=8 out of your patients as "few". Is approx. a thrid really "few"?

Author Response

I have some general comments:

  1. What is the real novelty of your study? What is the main aspect? I mean, the usage of ICG is not new. You also cite others adequately in that context. Since you do not have data about overall survival or disease recurrence (addressed in a large study, I understood) and you have also false positive events, I miss a bit the novelty, the additional information above "it just works"...
  2. For readers not familiar with minimal invasive settings, could you be so kind to explain a bit more what we see in the images. Label what is most important. Tell what are the numbers partially visible.  What is the T logo/label telling? Is there a scalebar/magnification value? etc.
  3. Could you add a statement on how you judge the false positive lesions? All imaging techniques have a risk of such events, for sure. Reasons are complex depending on technique, tissue(s), patient, contrast agents etc. What would you say (maybe on context to other techniques?), are these false positive things a problem? Or are they more like chance findings with a positive effect possible because another type of malign/benign malformation is detected? Or is more likely to be adverse, that you would take too much intervention with the corresponding risks?
  4. Small things:

Line 27 in Abstract: Fifry CRLMs ... It’s a typo, right?

Line 115 in label of Fig 1: One (a) is too much, right?

Line 152: You are stating n=8 out of your patients as "few". Is approx. a third really "few"?

Author Response

  1. In literature only small studies reported the use of this technique for the intraoperative identification of colorectal metastases during laparoscopic or robotic liver resection (Achterberg et al [12], Patel et al [14] and He et al [23]). The most important paper is a randomized controlled trial involving only 32 patients for each group. Therefore theknowledge about this topic is only preliminary.

The main points that make our study relevant especially for the clinical practice are: (a) The rim fluorescence pattern is typical of colorectal cancer. The majority of lesions had rim pattern (91.4%), and all lesions with rim fluorescence pattern were really metastases. However, what we didn’t expect to find was that three lesions with intraoperative partial or total fluorescence patterns were hepatocellular carcinoma (HCC) instead of colorectal cancers. (b) All patients with the integrity of the fluorescent rim around the CRLM had a radical resection (resection margin with a distal clearance >1 mm from the tumor, R0), while four patients (10.8%) with discontinuity of the rim fluorescence, protruding rim or residual rim patterns had positive resection margins (resection margins <1 mm from the tumor, R1). The typical rim is due to compressed hepatocytes, surrounding metastatic lesions, that have a decreased bile excretion ability. (c) We consider the rim fluorescence as the resection margin, therefore we believe that, in order to obtain a tumor-negative resection (R0), it is necessary to remove the entire fluorescent area with the metastasis included.

  1. We modified the figure legend.
  2. According to the literature, ICG retention can also occur in benign nodules, such as high-grade dysplasia, regenerative nodules, biliary hamartomas, and focal nodular hyperplasia. However we did not find any of these lesions at histopathological analysis. In our study the two false positive lesions were found in patients who underwent neoadjuvant chemotherapy. We considered these lesions as small subcapsular colorectal metastases (not identified preoperatively) with a complete histopathological response.
  3. We modified these errors.

Reviewer 2 Report

Comments and Suggestions for Authors

In this cohort study on 25 patients operated for colorectal lever metastasis the peroperative indocyanine green (ICG) fluorescence imaging was evaluated with the purpose to detect not recognized metastasis by pre- and peroperative imaging and a radical R0 resection. The manuscript is well written and easy to follow. 

Only patients with low or median disease burden were included and therefore information on the total number of patients operated with in the observation period would have been very informative. So please give us the number if available. 

It should appear from the title that the investigation was performed in a selected population wih mid of median disease burden. 

In 4 out of the 25 patients additional lesions (5 in total) were detected, but in only half of the patients malignancy could be proven by histology. At the same time, it is worthwhile to mention that only 34 of a total of the 50 metastases were recognizable with ICG alone. Do the authors have an explanation for this?

In the calculations of the sensitivity of the different imagining modalities it is presumed that the combination of laparoscopic ultrasonography (LUS) and ICG have detected alle possible metastasis. How sure can you be on this?

Were the aware of a possible R1 resection in patients with a “protruding rim” or “residual” rim at the ICG. Were any additional resection performed in these patients. If not – then why? If yes – what did histology of the resected margin show?

Author Response

In this cohort study on 25 patients operated for colorectal lever metastasis the preoperative indocyanine green (ICG) fluorescence imaging was evaluated with the purpose to detect not recognized metastasis by pre- and peroperative imaging and a radical R0 resection. The manuscript is well written and easy to follow.

  1. Only patients with low or median disease burden were included and therefore information on the total number of patients operated with in the observation period would have been very informative. So please give us the number if available.
  2. It should appear from the title that the investigation was performed in a selected population with mid of median disease burden.
  3. In 4 out of the 25 patients additional lesions (5 in total) were detected, but in only half of the patients malignancy could be proven by histology. At the same time, it is worthwhile to mention that only 34 of a total of the 50 metastases were recognizable with ICG alone. Do the authors have an explanation for this?
  4. In the calculations of the sensitivity of the different imagining modalities it is presumed that the combination of laparoscopic ultrasonography (LUS) and ICG have detected all possible metastasis. How sure can you be on this? Were the aware of a possible R1 resection in patients with a “protruding rim” or “residual” rim at the ICG. Were any additional resection performed in these patients. If not – then why? If yes – what did histology of the resected margin show?

Author Response

  1. In our Liver Unit (including only two surgeons), we performed approximately a total of 85 liver resections for colorectal metastases, including open, laparoscopic and robotic resections.
  2. We excluded patients with high burden disease requiring major liver resection, because ICG is a useful tool for parenchymal sparing liver resection and not so much for anatomical liver resection. The most important drawback of this technique is the depth of fluorescence penetration (up to 8 mm from the liver surface). Therefore in our study we included only patients with superficial colorectal cancer.
  3. In our study the two false positive lesions were found in patients who underwent neoadjuvant chemotherapy, therefore we believe that these lesions are small subcapsular colorectal metastases (not identified preoperatively) with a complete histopathological response.
  4. In patients with protruding rim and residual rim we performed an additional resection to obtain a radical resection (resection margin with a distal clearance >1 mm from the tumor). We consider the rim fluorescence as the resection margin and not as the tumor. The typical rim is due to compressed hepatocytes, surrounding metastatic lesions, that have a decreased bile excretion ability. Therefore we believe that, in order to obtain a tumor-negative resection, it is necessary to remove the entire fluorescent area with the metastasis included.

Reviewer 3 Report

Comments and Suggestions for Authors This paper summarizes the experience of the use of near infrared fluorescence imaging in combination with laparoscopic intraoperative ultrasound for the detection of colorectal liver metastases during minimally invasive liver resection surgeries. The authors acknowledge  the potential limitations of the study. However, in the light of overall limited data they they present some important findings that are of interest to many readers:
  1. Near infrared fluorescence imaging (NIRF) using indocyanine green (ICG) is a valuable tool in the detection and assessment of colorectal liver metastases (CRLMs) during minimally invasive liver resection. ​
  2. The combination of ICG fluorescence imaging and intraoperative ultrasound (IOUS) can improve the detection rate of superficial liver tumors. ​
  3. ICG fluorescence staining allows for the detection of additional small superficial lesions that may not be identified with other preoperative or intraoperative techniques. ​
  4. The integrity of the fluorescent rim around the CRLM is an indicator of a radical resection (R0), while the presence of a protruding or residual rim pattern may indicate positive resection margins (R1). ​
  5. ICG fluorescence imaging can be integrated with other conventional intraoperative imaging techniques to optimize intraoperative staging and improve surgical outcomes in patients with CRLMs. ​

Minor language editing is required but overall the study merits publication.

Comments on the Quality of English Language

Minor language editing is required but overall the study merits publication.

Author Response

This paper summarizes the experience of the use of near infrared fluorescence imaging in combination with laparoscopic intraoperative ultrasound for the detection of colorectal liver metastases during minimally invasive liver resection surgeries. The authors acknowledge  the potential limitations of the study. However, in the light of overall limited data they they present some important findings that are of interest to many readers:

  1. Near infrared fluorescence imaging (NIRF) using indocyanine green (ICG) is a valuable tool in the detection and assessment of colorectal liver metastases (CRLMs) during minimally invasive liver resection. ​
  2. The combination of ICG fluorescence imaging and intraoperative ultrasound (IOUS) can improve the detection rate of superficial liver tumours. ​
  3. ICG fluorescence staining allows for the detection of additional small superficial lesions that may not be identified with other preoperative or intraoperative techniques. ​
  4. The integrity of the fluorescent rim around the CRLM is an indicator of a radical resection (R0), while the presence of a protruding or residual rim pattern may indicate positive resection margins (R1). ​
  5. ICG fluorescence imaging can be integrated with other conventional intraoperative imaging techniques to optimize intraoperative staging and improve surgical outcomes in patients with CRLMs. ​

Minor language editing is required but overall the study merits publication.

Author Response

Thank you so much.

We improved the English language of the paper.

Round 2

Reviewer 1 Report

Comments and Suggestions for Authors

Thank you for your revision.

Sadly, I am not convinced by your revised manuscript. There is still the question about novelty (Question 1 in 1st report). The 2nd question or better the request for modifying your image description and labeling was not implemented.  My 3rd question for comments from your site about false positive results was not really addressed. You repeat what was already written, more or less. What about a judgement of your findings, a critical questioning and contextualizing of your observations also in context to other techniques? Is it acceptable to have such a amount of false positive results? Is it even good? Or would you expect negative adverse things based on these findings?

Author Response

  1. Thank you so much for the opportunity to discuss our manuscript with you. In the first question you ask us what's new about our manuscript. You stated that the use of ICG is not new. Yes, you're right, the use of ICG in hepatobiliary surgery has become increasingly widespread in the last years, but the majority of authors reported the use of ICG to identify segmental hepatic regions (positive and negative staining) during selective anatomical liver resection (segmentation), while we use ICG fluorescence as new toll for “parenchymal sparing” liver resections. We use ICG fluorescence as a visual feedback tool instated of the tactile feedback for the detection of sub-capsular lesions during minimally invasive liver resections.

In minimally invasive settings, the lack of tactile feedback on the hepatic surface, makes it challenging to detect sub-capsular CRLM with only ultrasound. Therefore, ICG fluorescence imaging can be integrated with conventional ultrasound to optimize intraoperative staging of superficial lesions. We believe that ICG fluorescence and intraoperative ultrasound can supplement each other. However, this technique is not useful for deep parenchymal tumours except those on the cut surface of the liver parenchyma. The most important drawback of this technique is the depth of fluorescence penetration (up to 8 mm -10 mm from the liver surface). We have admitted this limitations and inserted this sentence in the conclusion of the paper. As we have already answered previously, in literature only small studies reported the use of this technique for the intraoperative identification of colorectal metastases resection (Achterberg et al [12], Patel et al [14] and He et al [23]). I think that our experience can implement the evidences about this issue.

  1. We had already modified the figure legend.

Figure 3. ICG fluorescence staining allowed to detect: (a) the main nodule in S3 with partial fluorescence pattern; (b one additional smaller pericholecystic lesion with total fluorescence pattern.

Figure 4. Three cases of positive resection margins are shown (resection margins < 1 mm from the tumour): (a) discontinuity of the rim fluorescence; (b) residual rim patterns (bright fluorescent signal in the wound bed of the resection specimen); (c) protruding rim (fluorescent area protruding through the liver tissue of the resected specimen).

  1. In the second question you ask us what we think about false positive results. Yes, you're right, it is a problem. However, as already written previously, in our study the two false positive lesions were found in patients who underwent neoadjuvant chemotherapy. We considered these lesions as small subcapsular colorectal metastases (not identified preoperatively) with a complete histopathological response. On the other hand, ICG fluorescence staining allowed to detect 3 additional superficial lesions not identified with other preoperative or intraoperative techniques. Future research with larger scale multicentric study are needed to resolve these concerns.

Reviewer 2 Report

Comments and Suggestions for Authors

The authors have answered my questions sufficiently and have made the necessary corrections

Author Response

Thank you so much for the opportunity to discuss our manuscript with you